# Synergistic Effect of Two Plasticizers on Thermal Stability, Transparency, and Migration Resistance of Zinc Arginine Stabilized PVC

**DOI:** 10.3390/polym14214560

**Published:** 2022-10-27

**Authors:** Yanqin Shi, Yuchen Yao, Songyan Lu, Lukai Chen, Si Chen, Huiwen He, Meng Ma, Xu Wang

**Affiliations:** College of Materials Science and Engineering, Zhejiang University of Technology, Hangzhou 310014, China

**Keywords:** poly(vinyl chloride) (PVC), plasticizer, synergistic effect, thermal stability, transparency, migration resistance, Hansen solubility parameters theory

## Abstract

The effect of different plasticizers on thermal stability, transparency, and migration resistance of the PVC stabilized with zinc arginine [Zn(Arg)_2_] was investigated. The thermal stability, migration resistance, and transparency of PVC with tributyl citrate (TBC) were better than PVC with dioctyl phthalate (DOP) characterized by oven aging method, migration test, and near infrared-visible-ultraviolet spectrophotometer. At the same time, the longer the carbon chain in citric acid esters, the better the thermal stability and transparency of PVC sample. The hydroxyl group in citric acid esters is helpful to improve the thermal stability of PVC samples. However, the elongation at break and T_g_ value of PVC containing DOP were very close to those of PVC containing TBC. The calculation results of Hansen solubility parameters also illustrated that DOP had better compatibility with PVC than TBC. Therefore, the excellent transparency and thermal stability of TBC plasticized PVC were attributed to the good compatibility between TBC and Zn(Arg)_2_, which was verified by the solubility test. Lastly, the mixture of dioctyl terephthalate (DOTP) and TBC was used as plasticizers for Zn(Arg)_2_ stabilized PVC. When the ratio of TBC and DOTP was 1:1, the transparency, thermal stability, and migration resistance of Zn(Arg)_2_ stabilized PVC samples were better than those of PVC plasticized by DOP or TBC alone. The mechanism was that the compatibility between Zn(Arg)_2_ and PVC was greatly improved by the synergetic effect of TBC and DOTP, resulting in the improvement of thermal stability, migration resistance, and transparency of PVC samples.

## 1. Introduction

As one of the important polymers in the world, the annual output of polyvinyl Chloride (PVC) is second only to polyethylene (PE), and it is favored by people in the fields of automobile manufacturing, building decoration, children’s toys, and outdoor protective materials [1,2,3,4].Additionally, the PVC derivatives also have been used as active packaging materials [5,6]. The reason is that PVC has good mechanical properties, acid and alkali resistance, insulation, flame retardant, and so on. However, due to its special structure, it releases a large amount of HCl when the processing temperature is higher than 130 °C, leaving conjugated double bonds on the long chain of PVC, which makes PVC products yellow and seriously affects their performance [7]. The addition of thermal stabilizers can effectively ensure its structural stability, reduce the release of HCl, delay its color yellowing, and decrease the properties. Therefore, it is necessary to add thermal stabilizers when processing PVC [8,9]. Commonly used thermal stabilizers for PVC include lead salt [10], organotin [11], metal salt [12,13], organic thermal stabilizers [14]. At the same time, the plasticizer is another important processing additive for PVC. It enters between the molecular chains of PVC, and weakens the intermolecular force, resulting in the improvement of the thermal stability of PVC and the flexibility of PVC products. Most PVC products always contain thermal stabilizers and plasticizers at the same time.

PVC can be prepared into highly transparent samples by adding appropriate thermal stabilizers and plasticizers. Transparent PVC is widely used in medical materials, food packaging, and high-property building materials [2]. The thermal stabilizers and plasticizers for transparent PVC must have good compatibility with PVC [15]. Metal salt thermal stabilizers are recognized as the main environmental stabilizers for it has no heavy metal and proper thermal stability [16]. The thermal stability of metal salt thermal stabilizers is higher than that of organic thermal stabilizers [17], and the environmentally friendly is better than that of organotin [18]. However, the thermal stability efficiency of zinc stearate/calcium stearate (ZnSt_2_/CaSt_2_), which is the main metal salt thermal stabilizer commonly used in the market, is not high enough [18]. Many other auxiliary stabilizers need to be added with ZnSt_2_/CaSt_2_ to obtain excellent thermal stability, which leads to the poor transparency of PVC samples because of the poor compatibility of the auxiliary stabilizers and PVC. Thus, it is urgent for preparing another highly efficient and transparent thermal stabilizer for transparent PVC samples.

In the past years, a lot of new structures zinc organic thermal stabilizers, zinc maleate [19], zinc 11-maleimide [20], Zinc pentaerythritol [21], zinc mannitol complexes [22], and Zinc Orotate [23], had been designed and prepared for achieving the better thermal stability for PVC. Some of them had good long-term thermal stability, and some had good initial whiteness. However, no one could not give PVC outstanding thermal stability and transparency at the same time. In the past ten years, our research group has been committed to the research of thermal stabilizers for transparent PVC. Firstly, the thermal stability and transparency of PVC samples stabilized with a series of uracil derivatives [24,25,26,27] were studied. Through the calculation of Hansen solubility parameters, the relationship between the carbon chain length in bis-uracil derivatives and the thermal stability and transparency of PVC samples with bis-uracil derivatives was obtained [28]. Although alkyl-substituted bis(6-amino-1,3-dimethyluracil-5-yl) methane can be used as the main stabilizer, the whiteness retention time is still insufficient. Based on the research of organic thermal stabilizers, our group further designed and synthesized organic zinc thermal stabilizers, such as triazole based zinc containing complex [Zn (ttr) (St)] [29] and [Zn (ttr) (OAc)] [30] and obtained the rule of influence of the anionic structure in organic zinc thermal stabilizers on the thermal stability and transparency of PVC. At the same time, we found that the reason why [Zn (ttr) (St)] and [Zn (ttr) (OAc)] have excellent thermal stability is that they have a multi-step thermal stabilizer mechanism. Based on the research results of organic and organic zinc thermal stabilizers, we have successfully designed an organic zinc thermal stabilizer with excellent thermal stability and transparency on PVC. That is zinc arginine complex [Zn(Arg)_2_] [31]. The static thermal stability time is more than 180min. Importantly, the dynamic thermal stability of the PVC with Zn(Arg)_2_ is outstanding. That means Zn(Arg)_2_ is very promising to realize industrial applications.

The plasticizer is another main additive for PVC, which has a significant impact on the thermal stability and transparency of PVC samples [32]. Although many researchers have studied some plasticizers with different structures. There are including epoxidized soybean oil (ESBO), bis(2-ethylhexyl) adipate(DEHA), acetylated tributyl citrate (ATBC), glycerol acetate mixture(AGF), butylene terephthalate (PBAT) [33], polycinnamoyloxyethyl methacrylate-grafted cellulose nanocrystals (PCEM-g-CNCs) [34], methyl acetyl ricinoleate [35], bio-based polyurethane elastomer (BPU) [36], and epoxy p-phenylenedioleic acid ester (EPAE) [37]. Citrate plasticizer is a kind of environmentally friendly plasticizer. Its environmentally friendly properties and plasticizing properties have been confirmed. It is also one of the commercially available environmentally friendly plasticizers. However, the above research did not study the interaction between thermal stabilizer and plasticizer in the same PVC sample. Actually, both plasticizer and thermal stabilizer were added to transparent PVC at the same time [38]. Thus, it is very important to obtain plasticizers suitable for thermal stabilizers with specific structures for preparing high transparent and high thermal stability PVC samples. The plasticizer not only needs to have good compatibility with PVC so that it can enter between PVC molecular chains to play a good plasticizing role, but also needs to have good compatibility with the thermal stabilizer to improve the dispersion of the thermal stabilizer in PVC, so as to improve the thermal stability and transparency of PVC samples.

As a result, finding a plasticizer suitable for zinc arginine is conducive to improving the transparency and thermal stability of zinc arginine stabilized PVC samples. Although our previous studies have shown that zinc arginine has good thermal stability and transparency for PVC in the presence of DOP, the effect of plasticizer structure on thermal stability and transparency of zinc arginine stabilized PVC samples has not been investigated.

In this paper, the effects of DOP and four kinds of citric acid esters on the thermal stability, transparency, and migration resistance of Zn(Arg)_2_ stabilized PVC samples were studied. According to the experimental results and theoretical calculation, the mixture of two kinds of plasticizers were used to plasticize Zn(Arg)_2_ stabilized PVC samples at the same time, which significantly improved the thermal stability, transparency, and migration resistance of PVC samples. The synergistic effect of the two plasticizers was that the mixture of the two plasticizers enhanced the compatibility of Zn(Arg)_2_ and PVC. It provides a new way to obtain the plasticizers has synergistic effect with other thermal stabilizers for PVC.

## 2. Materials and Methods

### 2.1. Materials

Polyvinyl chloride (PVC, SG5) was produced by Xinjiang Tianye Group Co., Ltd. (Shihezi, China) Arginine (Arg, 98%) was supplied by Shanghai Macklin Biochemical Technology Co., Ltd. (Shanghai, China) Zinc acetate (Zn(Ac)_2_, 97%), Dioctyl terephthalate (DOTP, AR), dioctyl phthalate (DOP, AR), trimethyl citrate (TMC, AR), triethyl citrate (TEC, AR), acetyl triethyl citrate (ATEC, AR), tributyl citrate (TBC, AR), ethanol (99.5%) purchased from Aladdin Reagent (Shanghai, China) Co., Ltd. Zinc arginine [Zn(Arg)_2_] was produced according to our previous research [31].

### 2.2. Preparation of PVC Sheets

A 100phr PVC, 30phr plasticizer, and 2phr thermal stabilizer were mixed together and plasticized for 100 s on a two-roll mill (Labtech, Stockholm, Sweden, LRM-S-150/3E) at 150 °C with a front roll speed of 11.7 r/min and a rear roll speed of 9 r/min. The mixed PVC samples were pressed on an automatic tablet machine (Labtech Ltd., Stockholm, Sweden, LP-S-50) to form PVC sheets with the size of 2 mm × 3 cm × 3 cm. The pressing temperature was 150 °C and the pressure was 110 bar.

### 2.3. Evaluation of PVC Sheets

#### 2.3.1. Transparency

The transmittance curves of the PVC samples were obtained by a near-infrared/visible/ultraviolet spectrophotometer (Perkin Elmer, Waltham, MA, USA, Lambda 750S) in the wavelength range of 400 nm to 800 nm. The haze of the PVC samples was also tested using a near-infrared/visible/ultraviolet spectrophotometer (Perkin Elmer, Waltham, MA, USA, Lambda 750S) according to ASTM D-1003-00, which reflects the degree of light scattering of the PVC samples.

#### 2.3.2. Static Thermal Stability

The PVC sheets were placed in an oven at 180 °C and taken out for scanning every 10 min by a scanner (HP, Palo Alto, CA, USA, LaserJet Pro M329). The color changes of the PVC sheets were recorded until the PVC sheets were completely black.

#### 2.3.3. Mechanical Properties

The PVC samples with the size of 75 mm × 12 mm × 2 mm were also prepared by the two-roll mill and automatic tablet machine as same as the PVC sheets. The elongation at break of the PVC samples was tested by a testing machine (Instron, Norwood, MA, USA, 5966) according to the standard GB/T528-2009, where the tensile speed was 50 mm/min and the temperature was 23 ± 0.2 °C.

#### 2.3.4. Glass Transition Temperature

The samples for dynamic thermomechanical analysis were cut into 30 mm × 10 mm × 2 mm from the PVC sheets. A single cantilever test was carried out using dynamic thermomechanical analysis (DMA) (TA, New Castle, DE, USA, Q800) at a heating rate of 3 °C/min in the temperature range of −30 °C to 80 °C under 1 Hz frequency and 15 μm amplitude. The tanδ curves were recorded, and the glass transition temperatures was read.

#### 2.3.5. Migration Ratio of Zn(Arg)_2_

The PVC samples were taken out from the constant temperature and humidity box (Yiheng, China, BPS-50CL) at 50 °C and 50% humidity, then weighed (m_1_). Washed the thermal stabilizer migrated to the surface of the PVC sample with ethanol and dried it again in an oven at 50 °C for weighing again (m_2_). The migration ratio (R) of Zn(Arg)_2_ from the PVC sample was calculated as Equation (1).
(1)$$R=\frac{m_{1}-m_{2}}{m_{1}\times 0.015}\times 100\%$$

Meanwhile, the transmittance and haze of the PVC samples were also tested as stated in Section 2.3.1.

## 3. Results and Discussion

### 3.1. Effect of Citric Acid Esters on the Properties of PVC Stabilized by Zn(Arg)_2_

We found that the Zn(Arg)_2_ had excellent thermal stability for PVC in the previous research [31]. However, the synergetic effect between Zn(Arg)_2_ and plasticizers has not been studied. Firstly, DOP was used as the plasticizer for the PVC stabilized by Zn(Arg)_2_. The effect of the DOP content on the thermal stability and transparency of PVC was investigated, as shown in Figure 1 and Figure 2.

Figure 1 indicates that the PVC samples containing 30phr and 50phr DOP maintain excellent long-term thermal stability, and remain yellow-brown until 180 min. However, we found that the PVC samples with 50phr DOP turned yellow earlier than that with 30phr DOP, which may have relationship with the compatibility between Zn(Arg)_2_, DOP and PVC, resulting in the influence of the dispersion of Zn(Arg)_2_ in PVC matrix. Meanwhile, the transparency of PVC with 50phr DOP are also worse than that of PVC with 30phr DOP revealed in Figure 2, which has much lower transmittance and much higher haze. The results hint that the increase of the DOP content is not good for the thermal stability and transparency of PVC, although the more plasticizer, the less processing heat. It may be ascribed to the compatibility of DOP with Zn(Arg)_2_ or PVC.

For obtaining better thermal stability and transparency of PVC sample with Zn(Arg)_2_, citric acid ester was chosen as another kind of plasticizer, which is environmentally friendly with no biological toxicity, excellent compatibility, and high plasticizing efficiency, especially its excellent light resistance and cold resistance. Citric acid ester is always used in PVC products like children’s soft toys, medical products, cosmetics manufacturing, and other industries [39,40]. On the other hand, citric acid ester has stronger polarity than DOP, which has better compatibility with Zn(Arg)_2_, so it can improve the dispersion of Zn(Arg)_2_ in PVC matrix, leading to better thermal stability, transparency, and migration resistance of PVC samples. According to the different polarity, four kinds of citric acid ester were chosen to compound with Zn(Arg)_2_. They are trimethyl citrate (TMC), triethyl citrate (TEC), Acetyl triethyl citrate (ATEC), and tributyl citrate (TBC), which are listed in Figure 1. The structural difference between TMC, TEC, and TBC is the different length of the carbon chain. While ATEC has four acetyl groups, one more than the others. The PVC samples with 2phr Zn(Arg)_2_ and 30phr citric acid ester were prepared, and its thermal stability, transparency, tensile properties, and migration ratio were also studied. DOP was as the contrast.

Figure 3 shows the thermal stability results of PVC samples containing 2phr Zn(Arg)_2_ plasticized by 30phr plasticizer, respectively. It can be seen from the results that compared with DOP, PVC plasticized by TEC and TBC not only maintained an excellent long-term thermal stability time, more than 180 min, but also had an obvious improvement in the initial whiteness retention time. The PVC sample with DOP started to become dark yellow at about 70 min. Surprisingly, the time became yellow of PVC containing TEC and TBC were delayed to 140 min and 180 min, respectively. The thermal stability of the other PVC samples is worse. Although PVC with ATEC has the good initial whiteness like that with DOP, its initial whiteness retention time is poorer than the PVC containing DOP.

The difference in thermal stability between the PVC samples may have a relationship with the structures of plasticizers, which decides the compatibility of plasticizers with Zn(Arg)_2_ and PVC, leading to the different thermal stability effects of Zn(Arg)_2_ and plasticizing effect of the plasticizers. As we can see from Figure 3, PVC with TEC and TBC exhibited better thermal stability than PVC with TMC, indicating the citric acid ester with a longer carbon chain is benefit to the thermal stability of PVC stabilized with Zn(Arg)_2_. Meanwhile, the citric acid ester containing hydroxyl and ester groups (TEC and TBC) has better thermal stability effect on PVC than that only containing the ester group (ATEC). It may be due to the complexing ZnCl_2_ effect of hydroxyl groups [21].

For proving different effect of citric acid esters on the thermal stability of PVC stabilized by Zn(Arg)_2_ has related to the compatibility between plasticizer with Zn(Arg)_2_ or PVC, the transmittance and haze of the PVC samples were tested and displayed in Figure 4. It can be seen from the results that the PVC samples containing DOP, TEC, ATEC, and TBC have better transmittance and lower haze than that containing TMC. The transmittance is higher than 75% in the range of 400 nm to 800 nm. The PVC plasticized by ATEC and TBC have highest transmittance, reached 94% in the range of 500 nm to 800 nm. The haze of them is also lower than others, less than 15%. The transparency of the PVC samples indicate ATEC and TBC could disperse Zn(Arg)_2_ more homogeneous in PVC, or they have the better compatibility with PVC. At the same time, PVC with TMC showed the lowest transmittance and highest haze, and the transparency of PVC with TEC is between that of PVC with TBC and TMC. The results tell us that the long carbon train in plasticizer is benefit for improving the transparency of PVC stabilized with Zn(Arg)_2_, like the thermal stability showed in Figure 3.

The elongation at break of the PVC samples were also tested and shown in Figure 5. It is also found that the PVC with TBC has the highest elongation at break, reached 420%. The order of the elongation at break of PVC with TMC, TEC, and TBC is TMC < TEC < TBC, which is the same as transmittance and haze of these PVC samples. The possible mechanism is that the longer the carbon chain in citric acid ester, the better the compatibility between the citric acid ester with Zn(Arg)_2_ or PVC, which promoted the thermal stability of Zn(Arg)_2_ or enhanced the plasticizing effect. In addition, the ATEC also has high elongation at break, which is almost as same as the PVC with DOP, although the thermal stability of the PVC with ATEC is not good, as shown in Figure 3. It indicates the ester group may be beneficial to improve the plasticizing effect of plasticizer, but not good for thermal stability.

In order to further confirming the flexibility of the PVC chains in different samples, the glass transition temperatures (T_g_) of the PVC samples were tested and shown in Figure 6. Figure 6 shows that T_g_ of PVC with TBC is the lowest (41.91 °C), close to the T_g_ of DOP plasticized PVC sample (43.44 °C), which indicates that chains in the PVC sample with Zn(Arg)_2_ and TBC has good mobility. It may be caused by the little conjugated double bonds in PVC chains, or the small intermolecular force between PVC chains. The T_g_ of PVC with ATEC, TEC, and TMC groups were 51.80 °C, 54.81 °C, and 62.23 °C, respectively. The results also hint the longer the carbon chains in citric acid esters, the lower the T_g_ of PVC samples, and the ester group is benefit to the low T_g_, as same as the transparency and the elongation at break of the PVC samples.

For studying the compatibility between the different plasticizers and Zn(Arg)_2_, the migration ratio of the Zn(Arg)_2_ versus time in PVC samples was recorded, as shown in Figure 7. The migration ratio of Zn(Arg)_2_ in the TBC plasticized PVC sample is the lowest, with an initial migration ratio of 0.48%, and migration ratio after 40 days is only 1.20%. The migration resistance of Zn(Arg)_2_ in the TEC and ATEC plasticized PVC is also better than that of PVC with DOP. The migration ratio of Zn(Arg)_2_ in PVC with TEC and ATEC increased from 0.87% to 0.91%, and gradually increased to 1.50% and 1.56% in 40 days, while the migration ratio of Zn(Arg)_2_ in PVC with DOP varied from 1.18% at 0 days to 2.41% at 40 days. The migration ratio of Zn(Arg)_2_ in the PVC plasticized by TMC is the highest, rising from the initial 0.87% to 4.15%, which means that the compatibility between TMC and Zn(Arg)_2_ is poor. As a result, the compatibility between TBC and Zn(Arg)_2_ is best, leading to the low migration rate of Zn(Arg)_2_ in PVC.

As we know, the migration of Zn(Arg)_2_ affects the transparency of PVC samples, so the change of the average transmittance at 550–800 nm and haze was characterized over time. The results are shown in Figure 8.

As shown in Figure 8A, the transmittance of the two PVC samples with ATEC and TBC maintained a good transmittance (over 90%) for 40 days, which have the highest transmittance and a small decrease in transmittance, better than that of PVC with TEC and DOP. The average initial transmittance of them was 94.27% and 93.50%, which only decreased by 3.95% and 3.11%, respectively, after 40 days. Figure 8B show that, except for TMC, the haze growth trend of the PVC samples with citric acid esters is smaller than that of PVC with DOP, indicating that the three citric acid esters and Zn(Arg)_2_ have better compatibility in the PVC matrix. The mechanism may be that they has stronger polarity than DOP, which is close to the Zn(Arg)_2_. So the PVC with TEC, TBC, and ATEC have higher transmittance and lower haze. Among them, the initial haze of the TBC group was only 14.12%, and increased by 2.72% after 40 days, which indicating the TBC has the best compatibility with Zn(Arg)_2_. The results are consistent with the results of migration ratio test.

In conclusion, TBC plasticized PVC with Zn(Arg)_2_ has the best thermal stability, transparency, elongation at break, which were caused by the good compatibility between TBC and Zn(Arg)_2_, proved by the low T_g_ and excellent migration resistance.

### 3.2. Calculation of Compatibility between Plasticizers and PVC

The good compatibility between TBC and Zn(Arg)_2_ had been proved by the above studies. However, the compatibility between plasticizers and PVC has not been clear. So the Hansen solubility parameters (HSP) had been calculated.

HSP are essentially cohesive energy density, and is used to assess compatibility, affinity or dispersibility between substances. When the HSP values of two substances are closer, the compatibility between them is better [41]. The calculation formulas are shown in Equations (2)–(5).

Hansen solubility parameter:(2)$$\delta =\sqrt{\delta_{d}{}^{2}+\delta_{p}{}^{2}+\delta_{h}{}^{2}}$$

Dispersion Force Contribution in HSP:(3)$$\delta_{d}=\sum F_{di}/V$$

Polar force contribution in the HSP:(4)$$\delta_{p}=\sqrt{\sum F^{2}{}_{pi}\;}/V$$

Hydrogen Bond Contributions in the HSP:(5)$$\delta_{h}=\sqrt{(\sum E_{hi})/V\;}$$

The three parameters (*δ_d_*, *δ_p_*, and *δ_h_*) are the dispersion force, polar force, and hydrogen bonding contribution of the chemical structure to the HSP, respectively. *F_di_*, *F_pi_*, and *E_hi_* are the dispersion force, polar force, and hydrogen bond contribution of functional groups in the structure, respectively, and *V* is the molecular volume of the structure or group. The corresponding contribution values represented by different groups can be found in references, such as the Hansen Solubility Handbook [42], as shown in Table 1. Based on these values, the HSP value of a specific structure can be calculated [43,44,45]. The calculated values of the Hansen solubility parameters of citric acid esters are shown in Table 2 according to Equations (2)–(5).

Hansen also proposed the theory of Hansen sphere. In this theory, *δ_d_*, *δ_p_*, and *δ_h_* are listed on the x, y, z axes, respectively. The *δ* value of each chemical structure can find a specific point in the coordinate system. With the special point as the center, other substances within a certain radius have good compatibility with it. A Hansen solubility radius R_o_ can be obtained by computer simulation and calculating the interaction distance of the polymer [46,47,48,49,50]. So, we take the special point of PVC as the center of the circle and take 8.20 as the radius (R_o_). Thus, the Hansen sphere of PVC is obtained. At the same time, the points of the plasticizers in the coordinates are also calculated, and the distance (R_a_) from the point of plasticizer to the point of PVC is calculated according to Equation (6). The results also listed in Table 2, and the Hansen sphere is shown in Figure 9.
(6)$${\mathrm{Ra}}^{2}=4{\left(\delta_{d,p}-\delta_{d,s}\right)}^{2}+{\left(\delta_{p,p}-\delta_{p,s}\right)}^{2}+{\left(\delta_{h,p}-\delta_{h,S}\right)}^{2}$$
*p* represents the polymer solute and *s* represents the polymer solvent. Here, PVC is the solvent, and plasticizer is the solute. When the Ra/R_o_ value is within 1, the specific point of the plasticizer is in the Hansen sphere of PVC. It indicates that the plasticizer has good compatibility with PVC. Otherwise, when the Ra/Ro value is greater than 1, the specific point of the plasticizer is outside the Hansen sphere of PVC, indicates the poor compatibility between PVC and plasticizer.

Figure 9 displays that the Ra/R_o_ values of TBC, ATEC, and DOP were less than 1, which proves that they have excellent compatibility with the PVC matrix. The order of Ra/Ro values of TBC, ATEC, and DOP is ATEC < DOP < TBC, which means ATEC has the best compatibility with PVC, DOP is the second, and TBC is the worst. The order is not consistent with the transparency and T_g_ of the PVC with different citric acid esters. Moreover, the Ra/Ro values of TMC, TEC, and TBC also reveal that the longer the carbon chain in citric acid ester, the better the compatibility with PVC. The Ra/Ro value of ATEC is only 0.68, which hints that the acetyl group can promote the compatibility between citric acid ester and PVC.

In conclusion, the δ and Ra/Ro values of DOP is closer to that of PVC than TBC, which means the DOP has the best compatibility with PVC compared with the citric acid esters. However, the results of the transparency and the Tg test showed the PVC with TBC displayed the best transmittance and haze and the lowest Tg. The reason may have relationship with the compatibility between Zn(Arg)_2_ and plasticizer. That is to say, although TBC has poorer compatibility with PVC than DOP, it has better compatibility with Zn(Arg)_2_, which leads to the homogeneous dispersion of Zn(Arg)_2_, resulting in the excellent transparency, thermal stability, migration resistance, and low Tg. The results enlighten us if we want to get a PVC sample with highly thermal stability and transparency, the compatibility of plasticizer with thermal stabilizer and PVC should be both considered.

### 3.3. Synergistic Effect of Two Plasticizers for PVC Stabilized by Zn(Arg)_2_

In the above study, the excellent thermal stability, transparency, and migration resistance of PVC with TBC are ascribed to the good compatibility of TBC with Zn(Arg)_2_. The solubility of Zn(Arg)_2_ in TBC and DOP was tested, as shown in Table 3. The results verified that Zn(Arg)_2_ has better compatibility with TBC than DOP.

Whether PVC samples with better thermal stability and transparency can be obtained if two plasticizers (one has better compatibility with PVC and the other has better compatibility with thermal stabilizer) are used to plasticize Zn(Arg)_2_ stabilized PVC at the same time? So, we used the mixture of TBC and dioctyl terephthalate (DOTP) to plasticize Zn(Arg)_2_ stabilized PVC and tested the thermal stability, transparency, and migration resistance of the PVC samples. The total content of two plasticizers is still 30phr. The mass ratio between TBC and DOTP is 1:3, 1:1, and 3:1. DOTP is an isomer of DOP, which has the same *δ* and R_a_/R_o_ values as DOP. However, it is safer than DOP.

Figure 10 shows the thermal stability of PVC containing 2phr Zn(Arg)_2_ plasticized by DOTP and TBC in different proportions. It can be found that all the PVC samples maintained a long-term thermal stability time of more than 180 min. The initial whiteness of the PVC with TBC:DOTP = 1:1 is the best, although the whiteness retention time is not the best, which is only worse than the PVC with pure TBC, and better than other PVC samples. The results illustrate that there is a synergistic effect between TBC and DOTP when the mass ratio is 1:1, which can improve the compatibility of PVC and Zn(Arg)_2_, leading to improve the thermal stability of PVC.

Figure 11 shows the transmittance and haze of PVC with 2phr Zn(Arg)_2_ and 30phr the mixture of TBC and DOTP in different proportions, respectively. From the results, it was found that when TBC:DOTP = 1:1, the transmittance of the PVC sample is the highest and the haze is the lowest, which are 95.51% and 12.43%, respectively, better than those of the PVC with pure TBC (94.41% and 14.12%). The transmittance and haze of the PVC with pure DOTP are the worst, which are 92.74% and 15.25%, respectively, close to the PVC plasticized by DOP, as shown in Figure 4. The results are related to the similar structures of DOTP with DOP. Especially, it is found that the haze of three Zn(Arg)_2_ stabilized PVC samples with the mixture of TBC and DOTP is better than that of PVC with TBC or DOTP alone. It strongly proved that there is a synergistic effect between TBC and DOTP, which can promote the compatibility of thermal stabilizer Zn(Arg)_2_ with PVC and the plasticizing effect on PVC, resulting in the improvement of the transparency of PVC samples.

Figure 12 show the elongation at break of PVC samples plasticized in different ratios of TBC and DOTP containing 2phr Zn(Arg)_2_. As we can see, the elongation at break values of PVC samples with TBC and DOTP do not show a linear increase trend. There is a peak value of the PVC sample with TBC:DOTP = 1:1, the elongation at break reaches 354.5%. It is much better than that of PVC with pure DOTP, only 234.7%. The results mean TBC and DOTP with a specific ratio has a synergistic effect on enhance the thermal stability, transparency, and elongation at break. Otherwise, the elongation at break of PVC with pure TBC is the highest, reaching 420.0%, which is due to the TBC not only has the plasticization effect on PVC, but also has thermal stability effect, resulting in the PVC chains maintain good flexibility and have less degradation.

The plasticizing and thermal stability effects could be also reflected by the glass transition temperature tested by DMA, shown in Figure 13. The T_g_ of the PVC with TBC:DOTP=1:1 reaches 43.43 °C, which is lower than the other two PVC samples with the mixture of TBC and DOTP, and only little higher than the T_g_ of PVC with pure TBC (41.91 °C). The results also hint the synergetic effect of TBC and DOTP when the mass ratio of them is 1:1, which is consistent with the results of elongation at break test. The reason is also ascribed to that the mixture of TBC and DOTP has the excellent compatibility with Zn(Arg)_2_ and PVC, which enhances the thermal stability effect of Zn(Arg)_2_.

The migration ratios of Zn(Arg)_2_ in the PVC samples with the mixture of TBC and DOTP were also tested. The results are shown in Figure 14. When TBC:DOTP = 1:1, the migration ratio of Zn(Arg)_2_ from 0 day to 40 days are the lowest, only from 0.42% to 1.11%. It is lower than the migration of Zn(Arg)_2_ in PVC with pure TBC in the whole testing period. When the mass ratio of TBC and DOTP is 1:3 and 3:1, the migration ratios of the two PVC samples are much higher than that of PVC with pure TBC. The results strongly prove that Zn(Arg)_2_ has better compatibility with PVC in the existence of TBC and DOTP(1:1). It also illustrates that TBC and DOTP has obvious synergetic effect for enhancing the compatibility of PVC and Zn(Arg)_2_.

Figure 15 shows the change of transmittance and haze of PVC samples with different ratios of TBC and DOTP with time. When TBC:DOTP = 1:1, the PVC sample has the highest transmittance, which is about 95% at 0 days. The transmittance of PVC sample also kept at 92.61% after 40 days, only decreased by 2.5% compared with 0 days. At the same time, we found the order of the transmittance of the PVC samples is as same as the order of migration resistance. That is TBC:DOTP = 1:1 > TBC > TBC:DOTP = 3:1 > TBC:DOTP = 1:3 > DOTP. The results indicate the transmittance of the PVC sample is decided by the dispersion of Zn(Arg)_2_ in the PVC matrix. The more evenly Zn(Arg)_2_ is dispersed in the PVC matrix, the higher the transmittance of the PVC sample. Thus, the results also indicate that TBC and DOTP have an obvious synergistic effect, which promotes the dispersion of Zn(Arg)_2_ in PVC. Figure 15 also displays that the PVC sample with TBC:DOTP = 1:1 has the lowest haze, which is consistent with transmittance and migration resistance. The haze is only 12.43% at 0 days, and increased to 15.40% after 40 days, increasing by 2.97%.

Therefore, the key to the preparation of transparent PVC samples with high transparency lies in how to make the thermal stabilizer disperse evenly in the PVC matrix. This paper provides a method of using TBC and DOTP together, which realizes the uniform dispersion of Zn(Arg)_2_ in PVC matrix, and obtains a transparent PVC sample with excellent thermal stability, migration resistance, high transmittance, and low haze.

Based on the above data analysis, a mechanism of the synergetic effect of TBC and DOTP is proposed here: Zn(Arg)_2_ is a compound with strong polarity. TBC has stronger polarity than the DOTP. According to the similarity compatibility principle, TBC has better compatibility with Zn(Arg)_2_, and DOTP has better compatibility with the PVC matrix. So, when the TBC and DOTP are added in the ratio of 1:1, the Zn(Arg)_2_ and PVC achieve the best compatibility. Thus, the Zn(Arg)_2_ disperses homogeneously in the PVC matrix to achieve high thermal stability efficiency and transparency.

As a result, this work provides a new idea and method to prepare transparent PVC samples, that is using two kinds of plasticizers. One has good compatibility with thermal stabilizer, the other has good compatibility with PVC.

## 4. Conclusions

Firstly, the effects of four kinds of citric acid esters (TMC, TEC, ATEC, and TBC) on the thermal stability, transparency, and migration resistance of Zn(Arg)_2_ stabilized PVC samples were studied. The results show that TBC plasticized PVC sample not only has better thermal stability and transparency than DOP plasticized PVC sample, but also has low migration ratio of Zn(Arg)_2_ and T_g_. The reason is that TBC has better compatibility with Zn(Arg)_2_, leading to improve the dispersion of Zn(Arg)_2_ in PVC matrix. Hansen solubility parameters calculation results indicates that the compatibility of DOP with PVC is better than that of TBC. In order to further improve the dispersion of zinc arginine in PVC matrix, the mixture of DOTP and TBC were used to plasticize Zn(Arg)_2_ stabilized PVC samples. When the ratio of DOTP to TBC is 1:1, the thermal stability, transparency, elongation at break, T_g_, and migration resistance of PVC samples are better than those of PVC samples plasticized by TBC alone, indicating that the TBC and DOTP have significant synergistic effect. The thermal stability time of the PVC sample is longer than 180 min, the transmittance is as high as 94%, the haze is less than 13%, and the migration ratio within 40 days is less than 1%. The synergistic mechanism is that the mixture of TBC and DOTP make the excellent homogeneous dispersion of Zn(Arg)_2_ in the PVC matrix. This paper provides a new method to improve the thermal stability, transparency, and migration resistance of PVC samples, which is using the mixture of plasticizers. Some has good compatibility with PVC, and some has good compatibility with thermal stabilizer.

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
