# Peer review of "Synergistic Effect of Two Plasticizers on Thermal Stability, Transparency, and Migration Resistance of Zinc Arginine Stabilized PVC"

_polymers, 2022, doi:10.3390/polym14214560_

Round 1
Reviewer 1 Report
1. In many places' plural (figures) is used instead of singular figure.
2. How many specimens were used for transmittance and haze measurements? If it is done for more than one, standard deviation should be shown in the figures and tables.
3. How the thermal stability of the samples contributes to the improvement of elongation at break of the samples? These two are different type of tests, right? I think this explanation needs to revise.
4. From the DMA tests, extract the mechanical data and provide them in the manuscript. Also provide TGA (thermogravimetric analysis) data to support thermal stability of the samples.
Reviewer 2 Report
polymers-1960843-peer-review-v1
Synergistic Effect of Two Plasticizers on Thermal Stability, Transparency, and Migration Resistance of Zinc Arginine Stabilized PVC
Yanqin Shi, Yuchen Yao, Songyan Lu, Lukai Chen, Si Chen, Huiwen He, Meng Ma and Xu Wang
Dear Editor,
I looked at the manuscript. The Authors report a study of plasticizer effect on PVC. They report the effects of dioctyl phtalate and trimethyl citrate(TMC, AR), triethyl citrate, acetyl triethyl citrate, tributyl citrate, on the thermal stability, transparency, and migration resistance of zink arginine stabilized PVC samples. The results show that the longer the carbon chain in citric acid esters, the better the thermal stability and transparency of PVC sample. The work is interesting. The manuscript is well documented and well written. In the introduction section, the authors should mention about the PVC derivatives have been used as active packaging materials as reported in the following articles:
Food Chemistry 403 (2023) 134475. https://doi.org/10.1016/j.foodchem.2022.134475
Food Chemistry 344 (2021) 128644. https://doi.org/10.1016/j.foodchem.2022.134475
Kindest regards
Round 2
Reviewer 1 Report
The manuscript is revised carefully. However, I am not satisfied with the response to the comments.
1. Authors talk about the conjugate double bonds in PVC samples. How are these bonds developed in PVC?
2. If the TGA measurements are done in nitrogen atmosphere there won't be any problem for the machine. That was a lame excuse.
